# Near-Infrared Spectroscopy Study of Serpentine Minerals and Assignment of the OH Group

**Shaokun Wu [1], Mingyue He [1,*], Mei Yang [2], Biyao Zhang [1], Feng Wang [3] and Qianzhi Li [3]**

[1] Gemological Institute, China University of Geosciences, Beijing 100083, China; 3009210005@email.cugb.edu.cn (S.W.); zhangby889@163.com (B.Z.)
[2] Sciences Institute, China University of Geosciences, Beijing 100083, China; yangmei@cugb.edu.cn
[3] Shaanxi Mine in Hanyuan Jade Industry Co., Ltd., Hanzhong 723000, China; wang13891685963@163.com (F.W.); qianzhili2021@163.com (Q.L.)
* Correspondence: hemy@cugb.edu.cn

**Abstract:** Three different kinds of serpentine mineral samples were investigated using Fourier transform near-infrared spectroscopy (FTNIR). The results show that there are obvious differences in the characteristic infrared spectra of the three serpentine group minerals (lizardite, chrysotile, and antigorite), which can easily be used to identify these serpentine minerals. The combination of weak and strong peaks in the spectrum of lizardite appears at 3650 and 3690 cm$^{-1}$, while the intensities of the peaks at 4281 and 4301 cm$^{-1}$ (at 7233 and 7241 cm$^{-1}$, respectively) are similar. A combination of weak and strong peaks in chrysotile appears at 3648 and 3689 cm$^{-1}$ and at 4279 and 4302 cm$^{-1}$, and a single strong peak appears at 7233 cm$^{-1}$. In antigorite, there are strong single peaks at 3674, 4301, and 7231 cm$^{-1}$, and the remaining peaks are shoulder peaks or are not obvious. The structural OH mainly appears as characteristic peaks in four regions, 500–720, 3600–3750, 4000–4600, and 7000–7600 cm$^{-1}$, corresponding to the OH bending vibration, the OH stretching vibration, the OH secondary combination vibration, and the OH overtone vibration, respectively. In the combined frequency vibration region, the characteristic peak near 4300 cm$^{-1}$ is formed by the combination of the internal and external stretching vibrations and bending vibrations of the structural OH group. The overtone vibrations of the OH stretching vibration appear near 7200 cm$^{-1}$, and the practical factor is about 1.965. The near-infrared spectra of serpentine minerals are closely related to their structural differences and isomorphous substitutions. Therefore, near-infrared spectroscopy can be used to identify serpentine species and provides a basis for studies on the genesis and metallogenic environment of these minerals.

**Keywords:** near-infrared spectroscopy; lizardite; chrysotile; antigorite; OH group

## 1. Introduction

Infrared spectroscopy can be a powerful tool for identifying compounds, crystal structure and isomorphism. In geology, mid-infrared and near-infrared spectroscopy are commonly used for research. Mid-infrared (MIR) generally refers to the infrared spectrum in the range 400–4000 cm$^{-1}$ (25–2.5 μm). Among them, the 1500–4000 cm$^{-1}$ region can identify the characteristic fundamental frequency vibration of some groups. The 400–1500 cm$^{-1}$ region is called the "fingerprint area" [1], which can identify specific molecular structure from their spectrum. The near-infrared spectrometer was available as early as the 1920s, long before the popularization of the mid-infrared spectrometer laboratory [2]. Near-infrared (NIR) generally refers to the infrared spectrum in the range 4000–12,500 cm$^{-1}$ (2.5–1 μm). It has been widely used in agriculture, the chemical industry, and other fields due to its fast and non-destructive characteristics [3–6]. In geology, near-infrared spectroscopy mainly reflects the combination bands and overtone bands of water or different functional groups in minerals structure, as well as the combination modes of the hydroxyl group and metal ions, which can reflect the differences in the composition

and structure of minerals. Therefore, it is very useful for the identification and study of minerals.

Serpentine minerals are trioctahedral phyllosilicates type 1:1 and consist of $SiO_4$ tetrahedral sheets and octahedral brucite-like sheets. They produce different structures because of the lattice mismatch between the octahedral (O) and the tetrahedral (T) sheets. The compensation for this mismatch occurs through chemistry changes or through curvature of the layers [7]. Based on these different structures, serpentine can be divided into three types: lizardite, antigorite, and chrysotile. Lizardite has a flat crystal structure, where the planar sheet presents an ideal layer topology. Antigorite shows a corrugated wavy layer characterized by a modulated structure. Chrysotile with the cylindrical spiral wrapping of the 1:1 layer exhibits a fibrous structure. Naturally occurring intergrowths of the three types of serpentine minerals are very common. Among the conventional techniques used to identify serpentine minerals without losing microstructural information, only Transmission Electron Microscopy with EDS (TEM-EDS) gives unambiguous results; however, sample preparation is complex and time-consuming, and the interpretation of the electron diffraction patterns is not easy [8]. Therefore, it is necessary to find a simple characterization technique to identify and distinguish between these three minerals.

Infrared spectroscopy is a powerful tool for quickly identifying different types of clay minerals. The IR spectra of phyllosilicates have been summarized by Farmer [9]. Serpentines exhibit $\nu OH$ (OH-stretching) bands in the range 3650–3700 cm$^{-1}$ and $\delta OH$ (OH-vibrating) bands near 618 and 646 cm$^{-1}$, due to their inner and surface hydroxyls [9]; however, there is no specific distinction between the peaks of the inner and surface hydroxyl groups. Cheng et al. [10] measured and distinguished five phyllosilicate minerals using near-infrared spectroscopy, including pyrophyllite, muscovite, talc, chrysotile, and kaolinite. Li et al. [11] analyzed the near-infrared spectra of different types of silicate minerals and concluded that there were different types of combined frequency and double-frequency peaks for water in gem minerals. Zhang et al. [12] compared TO- and TOT-type layered silicates and found that layered silicates have a long hydrogen bond, which widens the stretching vibration band of O-H, shifts the band in the direction of the long wave, and sometimes causes double-peak phenomenon to appear. The stretching frequency of the hydroxyl group is also affected by the octahedral cation coordinated with the hydroxyl group.

The purpose of this study is to demonstrate the differences in the near-infrared spectra of three types of serpentines and to discuss the assignments of the major near-infrared bands, which can be used to quickly identify and distinguish between serpentine group minerals. The results of this study are applicable to the identification and research of jewelry, jade, and cultural relics. In addition, the attribution of the OH group's vibration spectrum peak is discussed in detail, which is applicable to investigations of the metallogenic environments and compositional differences of serpentine. Finally, careful measurements and thoughtful interpretations of the infrared spectra of clay minerals may be critical to the interpretation of remotely sensed data from the Earth and Mars.

## 2. Materials and Methods

The experimental samples were divided into three groups according to their mineral type. The lizardite and chrysotile (asbestos) samples were from the Hanzhong area, Shaanxi Province, China; the antigorite samples were from the Xiuyan area, Liaoning Province, China. The refractive index of each sample was between 1.55 and 1.56; the specific gravity was between 2.56 and 2.61. Typical mineral samples were selected from the three groups for this study. The uniform part of each sample, without other phases, was used for the experiments.

SEM-EDS analyses of the serpentine samples before this study showed all samples contain the isomorphic element iron, which is very common in serpentine minerals. There are few other kinds of isomorphic elements (Table 1).

**Table 1.** Mineral composition of the samples.

| Samples | Minerals | Form | Color | Isomorphism |
|---|---|---|---|---|
| DWH14-lz | | Block | Yellow | Fe/0.7%, Al/0.1% |
| ZH41-lz | | Block | Brownish yellow | Fe/0.9% |
| Z3-lz | Lizardite | Block | Brown | Fe/1.0%, Mn/0.1%, Al/0.1% |
| SZ17-lz | | Block | Dark brown | Fe/0.8%, Mn/0.2%, Al/0.1% |
| SM-ctl | Chrysotile | Fiber | Brown | Fe/0.7%, Mn/0.1%, Al/0.2% |
| H7-atg | Antigorite | Block | Light yellow | Fe/0.4%, Al/0.1% |
| H11-atg | | Block | Colorless | Fe/0.3% |

The X-ray diffraction (XRD) data were acquired using the SmartLab X-ray powder diffractometer at the Institute of Earth Science, China University of Geosciences, Beijing (CUGB). The system was equipped with a conventional copper target X-ray tube (set to 45 kV and 200 mA) and a graphite monochromator, with a stepping-scanning mode with a scanning speed of 4°/min and a step length of 0.02° in the range 3–90°. The testing temperature was 15 °C and the humidity was 22%. The samples were pulverized to 200 mesh powders using an agate mortar and stored immediately in a plastic bag to minimize contamination and oxidation. The results were normalized and analyzed using the MDI Jade 6.5 software and the International Center of Diffraction Database (ICDD).

The Fourier transform near-infrared spectroscopy (FTIR) measurements were performed using the Bruker Tensor II spectrometer at the National Infrastructure of Mineral, Rock, and Fossil Resources for Science and Technology (NIMRF). The spectra were collected in the frequency ranges of 400–4000 cm$^{-1}$ (MIR) and 4000–10,000 cm$^{-1}$ (NIR) using the KBr compression transmission method, with a resolution of 4 cm$^{-1}$, and 64 scans were accumulated to improve the signal to noise ratio. The testing temperature range was 10–23 °C, and the humidity was less than 22%. The NIR data were processed by baseline removal and normalization.

The regions of 520–700 cm$^{-1}$ and 3600–3800 cm$^{-1}$ in the mid-infrared (MIR) were complex and the PeakFit v4.12 software was used to fit the peaks in order to find the exact locations of their component peaks. All of the spectra were fitted using a combined Gauss–Lorentz Area function, with r$^2$ > 0.99.

## 3. Results

### 3.1. X-ray Diffraction

The XRD patterns of the seven selected serpentine minerals with standard XRD patterns are shown in Figure 1. The samples DWH14-lz, ZH-41-lz, Z-3-lz, and SZ-17-lz have patterns identical to that of PDF 86-0403, strong peaks at d$_{(001)}$ = 12.079–12.099° and d$_{(002)}$ = 24.301–24.340° and medium intensity peaks at d$_{(-1-11)}$ = 35.861–35.919°, which is the characteristic pattern of lizardite-1T. The sample SM-ctl has a pattern identical to that of PDF 27-1275, showing strong peaks at d$_{(002)}$ = 12.020° and d$_{(004)}$ = 24.300°, which is the characteristic pattern of chrysotile. The pattern has fewer but wider peaks and the shape of the peak at d$_{(020)}$ = 19.500° is asymmetric. The samples H7-atg and H11-atg have patterns identical to that of PDF 07-0417, which is the characteristic pattern of antigorite. The patterns are clear and strong (the weak peak is not obvious after normalization), with strong peaks at d$_{(001)}$ = 12.159–12.160° and d$_{(002)}$ = 24.580–24.599°. There is a double peak near d$_{(-131)}$ = 35.500–35.520° and d$_{(211)}$ = 37.022–37.043°. All of the samples were found to be almost pure minerals without any other phases.

### 3.2. Characteristics of MIR

In the mid-infrared range, there are four main spectral bands: 400–500 cm$^{-1}$, 520–700 cm$^{-1}$, 850–1100 cm$^{-1}$, and 3600–3800 cm$^{-1}$. The position, intensity, and number of spectral peaks are very consistent with the standard spectra of serpentine minerals (Figure 2), and no additional peaks appear, indicating that all of the samples are mainly composed of serpentine minerals.

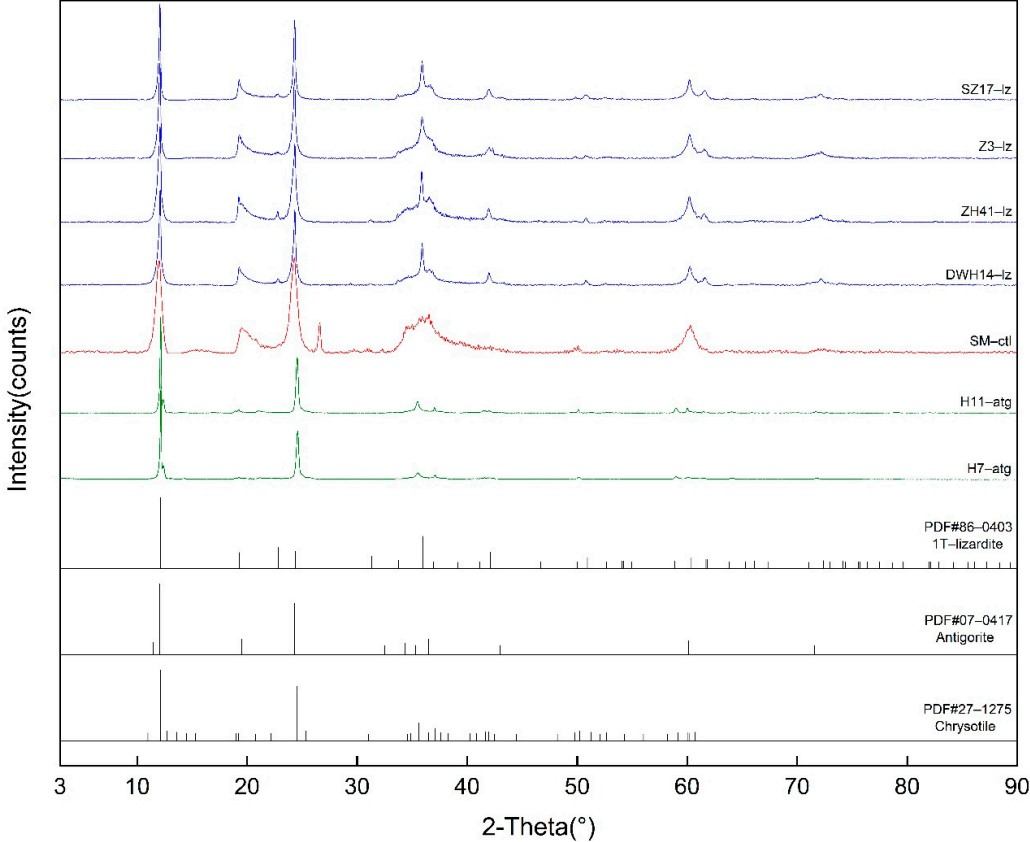

**Figure 1.** Powder XRD patterns of the samples.

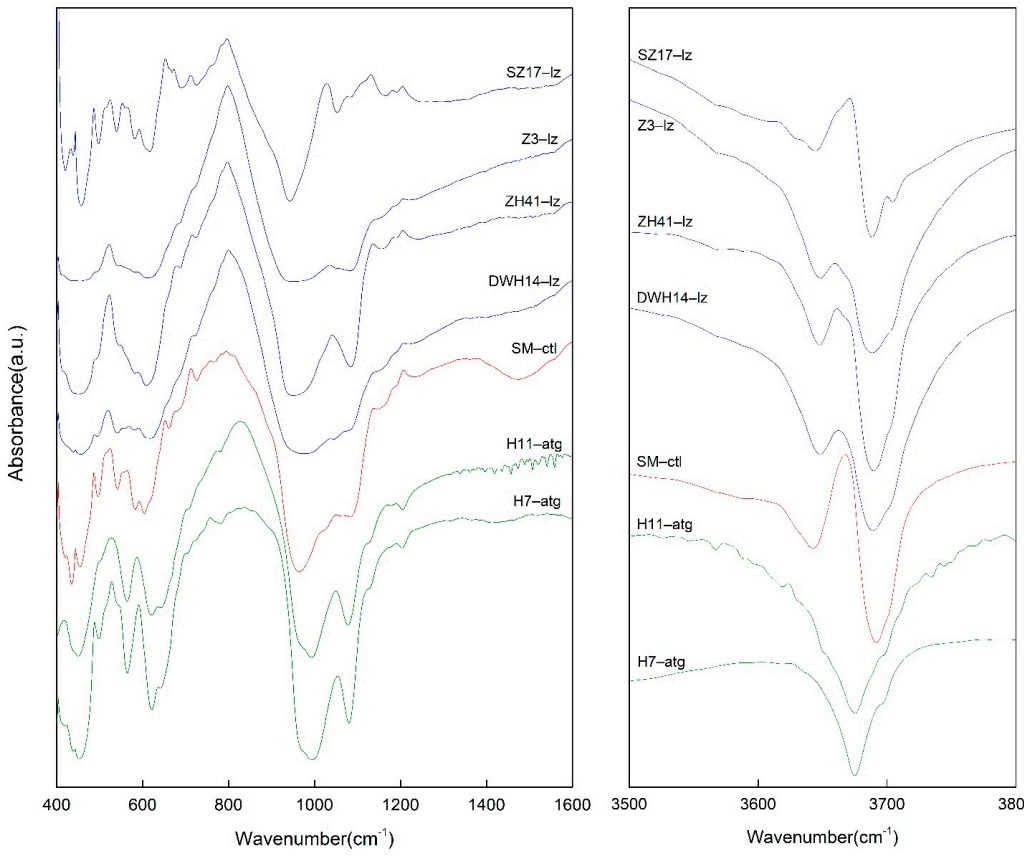

**Figure 2.** MIR spectra of the serpentine samples.

There are five peaks in the range 520–700 cm$^{-1}$. The spectra of DWH14-lz, ZH41-lz, Z3-lz, SZ17-lz and SM-ctl are similar, with two peaks related to SiO$_4$ tetrahedral deformation vibration near 550 and 580 cm$^{-1}$ and the strongest peak near 609 cm$^{-1}$. There is only one absorption peak at 564 cm$^{-1}$ for H7-atg and H11-atg, and there is a new peak at 666 cm$^{-1}$, which is different from the other five samples. The bending and translation vibration peaks of OH in antigorite are generally four to ten wavenumbers lower than those for lizardite and chrysotile.

The peaks in the range of 3600–3800 cm$^{-1}$ are usually attributed to OH stretching vibrations. The outer sloping OH stretching vibration of the serpentine minerals is at about 3650 cm$^{-1}$ and the vertical stretching vibration of the outer OH group is around 3670 and 3690 cm$^{-1}$ [13–16]. Antigorite has a unique small peak at 3632 cm$^{-1}$ and the stretching vibration of the inner OH group near 3700 cm$^{-1}$ is not present. This may be due to the periodically symmetric structure, which causes the inner OH vibrations to cancel each other.

Bishop [17] reported that the center of the OH bending vibrations were located at 606 cm$^{-1}$ for chrysotile and at 625 cm$^{-1}$ for lizardite. After fitting the peaks, it was found that the vibration peaks of the OH groups in the three types of serpentines were similar, with peaks at 609, 630, and 649 cm$^{-1}$, but their strengths varied significantly. Figure 3 shows DWH14-lz, SM-ctl, and H7-atg as representative examples (the other samples have similar patterns). The inner OH bending vibration centers of lizardite and chrysotile are around 609 cm$^{-1}$, 630 cm$^{-1}$, and 649 cm$^{-1}$, which are assigned to the OH translational vibration mode. In the range of OH stretching, for antigorite, the peak at 3674 cm$^{-1}$ is stronger than the peak at 3693 cm$^{-1}$, which are both assigned to the outer OH vertical stretching vibration. The opposite relationship occurs for those peaks in lizardite and chrysotile. Antigorite also lacks the peak near 3700 cm$^{-1}$ that is assigned to the inner OH stretching vibration. Figure 4 shows DWH14-lz, SM-ctl, and H7-atg as representative examples (the other samples have similar patterns). The detailed peak positions and band assignments of the samples are presented in Table 2.

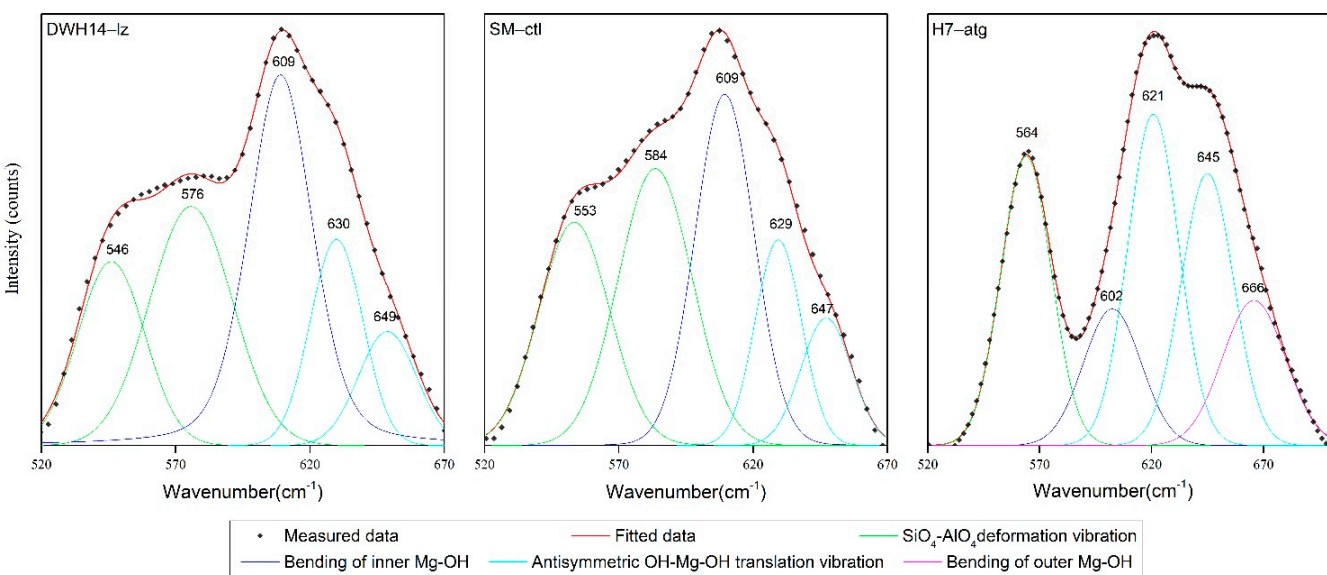

**Figure 3.** Band component analysis of the 520–700 cm$^{-1}$ region with DWH14-lz as example of lizardite, SM-ctl as example of chrysotile, and H7-atg as example of antigorite.

### 3.3. Characteristics of NIR

Serpentine minerals have multiple secondary combination bands in the range 4000–4500 cm$^{-1}$, with the strongest peak near 4280–4300 cm$^{-1}$ (Figure 5a). The band centers are listed in Table 3. The results show that lizardite exhibits equal-strength double

peaks near 4280 and 4301 cm$^{-1}$. The 4304 cm$^{-1}$ peak of chrysotile is stronger, and the 4279 cm$^{-1}$ peak appears as a shoulder. Antigorite has a weaker shoulder peak at 4279 cm$^{-1}$, and a new shoulder peak appears at 4315 cm$^{-1}$.

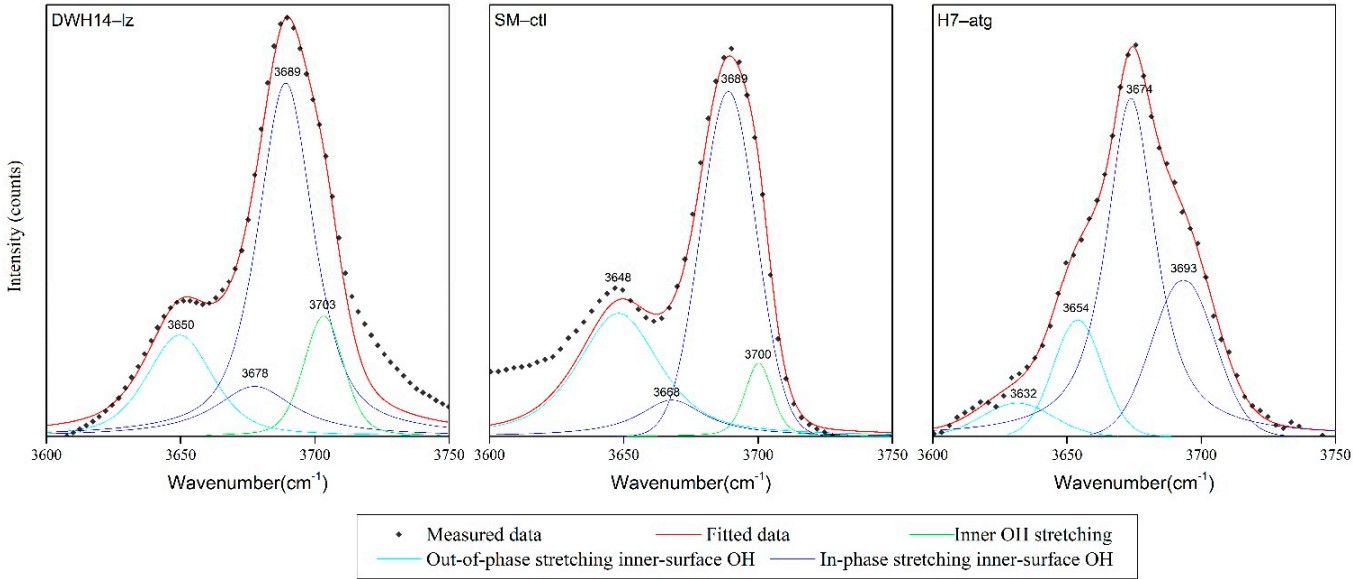

**Figure 4.** Band component analysis of the 3600–3750 cm$^{-1}$ region with DWH14-lz as example of lizardite, SM-ctl as example of chrysotile, and H7-atg as example of antigorite.

**Table 2.** The MIR bands related to the serpentine samples and their assignments (in cm$^{-1}$).

| Band Assignment [14,18–23] | DWH14-lz | ZH41-lz | Z3-lz | SZ17-lz | SM-ctl | H7-atg | H11-atg |
|---|---|---|---|---|---|---|---|
| Antisymmetric Mg-OH translation | 441 | 441 | 438 | 438 | 435 | 436 | 435 |
| Mg-OH translation + $\upsilon_6$(e) SiO$_4$ | 462 | 462 | 463 | 463 | 454 | 449 | 449 |
| SiO$_4$-AlO$_4$ deformation vibration | 546 | 553 | 551 | 553 | 553 | 564 | 564 |
| | 576 | 580 | 580 | 583 | 584 | | |
| Bending of inner Mg-OH | 609 | 609 | 609 | 610 | 609 | 602 | 600 |
| Antisymmetric OH-Mg-OH | 630 | 630 | 629 | 630 | 629 | 621 | 619 |
| translation vibration | 649 | 648 | 646 | 649 | 647 | 645 | 644 |
| Bending of outer Mg-OH | | | | | | 666 | 666 |
| Si-O stretching vibration | 961 | 962 | 963 | 956 | 963 | 970 | 969 |
| Si-O$_b$-Si stretching vibration | 1022 | 1022 | 1024 | 1018 | 1026 | 994 | 993 |
| Si-O$_{nb}$ stretching vibration | 1080 | 1078 | 1080 | 1079 | 1080 | 1080 | 1077 |
| Outer OH sloping stretching vibration | 3650 | 3647 | 3649 | 3648 | 3648 | 3632 | 3632 |
| | | | | | | 3654 | 3653 |
| Outer OH vertical stretching vibration | 3678 | 3665w | 3670w | 3667 | 3668 | 3674 | 3674 |
| | 3689 | 3688 | 3688 | 3688 | 3689 | 3693 | 3696 |
| Inner OH stretching vibration | 3703 | 3704 | 3705 | 3705 | 3700 | | |

All of the spectra have a large number of disordered but clear peaks in the ranges 5100–5500 cm$^{-1}$ and 7000–7500 cm$^{-1}$ (Figure 5b). Bishop [24] suggested that these bands are probably due to a small amount of H$_2$O molecules trapped at grain boundaries or are associated with impurities in the samples. However, our experimental results show that the peak position does not shift due to the influence of the type of serpentine and contents of the impurities. Thus, it is unlikely to be caused by isomorphism and is likely related to the water in the crystal structure. Cheng [10] pointed out that the unbound water in the interlayer structure of clay minerals causes a series of absorption peaks near 5260 cm$^{-1}$ and 7150 cm$^{-1}$, which are generally not assigned. There is a broad absorption band at 5000–5300 cm$^{-1}$ in chrysotile, which is reported to be caused by a large amount of adsorbed

water in the tubular structure. A similar broad absorption band was observed in the range of 3300–3500 cm$^{-1}$.

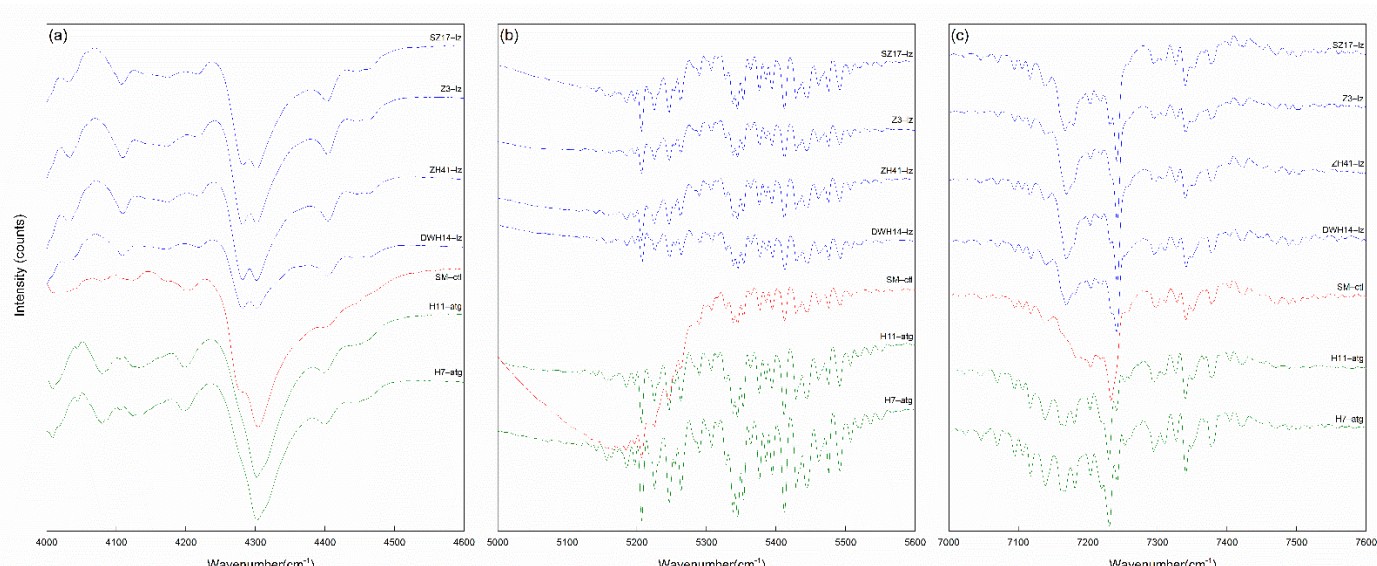

**Figure 5.** NIR spectra of the serpentine samples. (**a**) Range of the OH-combination bands (4000–4600 cm$^{-1}$); (**b**) adsorbed water bands (5000–5600 cm$^{-1}$); (**c**) range of the OH-stretching overtone bands (7000–7600 cm$^{-1}$).

**Table 3.** The bands in the range 4000–4600 cm$^{-1}$ for the related samples (in cm$^{-1}$).

| J. L. Post [18] | Baron [19] | DWH14-lz | ZH41-lz | Z3-lz | SZ17-lz | SM-ctl | H7-atg | H11-atg |
|---|---|---|---|---|---|---|---|---|
| 4010 | 4001 | 4009 | 4009 | 4009 | 4009 | 4009 | 4009 | 4009 |
| | | | | | | | 4044 | 4043 |
| 4076 | 4077 | | | | | 4078 | 4079 | 4079 |
| 4123 | 4121 | 4109 | 4109 | 4109 | 4108 | 4106 | 4105 | 4105 |
| 4196 | 4199 | 4218 | 4218 | 4218 | 4218 | 4201 | 4197 | 4197 |
| 4279 | 4274 | 4281 | 4281 | 4281 | 4281 | 4279 | 4279 | 4279 |
| 4303 | 4307 | 4301 | 4301 | 4301 | 4301 | 4304 | 4302 | 4301 |
| 4315 | | | | | | | 4315 | 4315 |
| 4401 | 4409 | 4404 | 4404 | 4404 | 4404 | 4401 | 4400 | 4400 |

The first fundamental overtone of the OH stretching vibrations of layered silicate is in the range 7160–7300 cm$^{-1}$ (Figure 5c). The statistics and assignments of the peaks are presented in Table 4. The weak interlayer water peaks were excluded. There are two strong peaks at 7170 and 7240 cm$^{-1}$ in the lizardite spectra. Chrysotile only has a strong peak at 7233 cm$^{-1}$, a weak shoulder peak at 7234 cm$^{-1}$, and very weak peak at 7170 cm$^{-1}$. Antigorite has a strongest peak at 7231 cm$^{-1}$ and a clear peak at 7243 cm$^{-1}$. It is not accurate to judge the existence of chrysotile solely on whether there is strip near 7233 cm$^{-1}$ [25].

**Table 4.** The bands in the range 7000–7600 cm$^{-1}$ for the related samples (in cm$^{-1}$).

| Outer OH | DWH14-lz | ZH41-lz | Z3-lz | SZ17-lz | SM-ctl | H7-atg | H11-atg |
|---|---|---|---|---|---|---|---|
| $\nu_{OH\text{-}s}$ | 7170 | 7168 | 7170 | 7167 | 7170 | 7168 | 7167 |
| $\nu_{OH\text{-}s}$ + $\nu_{OH\text{-}v}$ | 7204 | 7203 | 7204 | 7204 | 7204 | 7204 | 7203 |
| $\nu_{OH\text{-}v}$ | 7234 | 7233 | 7233 | 7231 | 7233 | 7231 | 7230 |
| $\nu_{OH\text{-}v}$ | 7241 | 7241 | 7242 | 7241 | 7243 | 7243 | 7241 |

## 4. Discussion

The band frequency relationships between the MIR and NIR spectra were determined via trial-and-error summations. Post [15] pointed out that there are two theories regarding the assignment of the peak at 4300 cm$^{-1}$. The stretching vibrations of the inner OH and outer OH are combined with different OH bending vibrations [18,26,27]. However, according to Bishop [17], peaks at 4280 and 4300 cm$^{-1}$ correspond to the combination of the internal or external hydroxyl stretching vibration and the bending vibration of the same hydroxyl group, and the peak at 4315 cm$^{-1}$ has not been analyzed. Combined with the data obtained in our experiment, we determined a more suitable peak assignment: the absorption band at 4280 cm$^{-1}$ was attributed to the combination of the 3650 cm$^{-1}$ assigned outer OH sloping stretching vibration and the 630 cm$^{-1}$ assigned antisymmetric OH-Mg-OH translation vibration. Additionally, the band at 4300 cm$^{-1}$ was attributed to the combination of the 3690 cm$^{-1}$ assigned outer OH sloping stretching vibration and the 610 cm$^{-1}$ assigned antisymmetric OH-Mg-OH translation vibration. The specific corresponding values of the three serpentine minerals are shown in Table 5. The outer OH sloping stretching vibration of H7-atg and H11-atg participates in the combination as a whole and the average value of 3643 cm$^{-1}$ has good results. All of the errors are within 10 cm$^{-1}$ [28]. The peak at 4279 cm$^{-1}$ is not obvious due to the weak OH sloping stretching vibration in antigorite. The combination of 3674 cm$^{-1}$ assigned to the outer OH vertical stretching vibration, with 620 cm$^{-1}$ assigned to the antisymmetric OH-Mg-OH translation vibration, is strong and clear. The 3674 cm$^{-1}$ peak also combines with the 645 cm$^{-1}$ peak assigned to the antisymmetric OH-Mg-OH translation vibration, presenting as a shoulder peak at 4315 cm$^{-1}$. In the other two kinds of serpentines, the peak intensity at 648 cm$^{-1}$ is weak, so the 4315 cm$^{-1}$ peak is difficult to observe. All of the samples exhibit an obvious peak near 4400 cm$^{-1}$. Bishop [24] believes that $Fe^{3+}$ replaces Mg and that the formation of $Fe^{3+}Mg_2$-OH causes this peak, which demonstrates that the substitution of Fe for Mg in the octahedral layer is common in serpentine.

The OH group vibration of serpentine in the mid-infrared region corresponds to the first fundamental overtone of the OH group stretching vibrations in the near-infrared range. The specific corresponding values of the three types of serpentines are presented in Table 6. The peak near 7170 cm$^{-1}$ belongs to the overtone of the outer OH sloping stretching vibration near 3648 cm$^{-1}$. The peak near 7204 cm$^{-1}$ belongs to the overtone of the combination of the outer OH sloping and vertical stretching vibrations near 3648 and 3689 cm$^{-1}$. The peaks near 7233 and 7241 cm$^{-1}$ belong to the overtone of the outer OH vertical stretching vibration. The outer OH sloping stretching vibration of antigorite also participates in the overtone as a whole.

The overtone of the vertical stretching vibration of the inner OH group should be around 7273 cm$^{-1}$, but there is no absorption in the spectrum at this position. It is possible that the vibrating direction of the OH group is limited, and thus the overtone vibrations cancel each other.

Due to the non-ideality of the structure and composition, the average factor between the OH stretching vibration and its overtone is 1.965 (+0.0026, −0.0037) rather than 2, which is consistent with Bishop [17] but more accurate. Among the samples, the average factor of samples from the Hanzhong area is less than 1.965, while the average factor of samples from the Xiuyan area is greater than 1.965, which may be used as the basis for origin differentiation, although future experiments with more samples will be required to confirm.

**Table 5.** The major NIR bands in the range 4000–4600 cm$^{-1}$ related to the serpentine samples and their corresponding MIR peaks (in cm$^{-1}$).

| | Lizardite | | | | | Chrysotile | | | | | Antigorite | | | |
|---|---|---|---|---|---|---|---|---|---|---|---|---|---|---|
| | Measured Peak | Fundamental Peaks | Theoretical Peak | Δ | | Measured Peak | Fundamental Peaks | Theoretical Peak | Δ | | Measured Peak | Fundamental Peaks | Theoretical Peak | Δ |
| DWH14-lz | 4281 | 3650 + 630 | 4280 | 1 | SM-ctl | 4279 | 3648 + 629 | 4277 | 2 | H7-atg | 4279 | 3643 * + 645 | 4288 | 9 |
| | 4301 | 3689 + 609 | 4298 | 3 | | 4304 | 3689 + 609 | 4298 | 6 | | 4302 | 3674 + 621 | 4295 | 7 |
| ZH41-lz | 4281 | 3647 + 630 | 4277 | 4 | | | | | | | 4315 | 3674 + 645 | 4319 | 4 |
| | 4301 | 3688 + 609 | 4297 | 4 | | | | | | H11-atg | 4279 | 3643 * + 644 | 4288 | 9 |
| Z3-lz | 4281 | 3649 + 629 | 4278 | 3 | | | | | | | 4301 | 3674 + 619 | 4295 | 6 |
| | 4301 | 3688 + 609 | 4297 | 4 | | | | | | | 4315 | 3674 + 644 | 4319 | 4 |
| SZ17-lz | 4281 | 3648 + 630 | 4278 | 3 | | | | | | | | | | |
| | 4301 | 3688 + 610 | 4298 | 3 | | | | | | | | | | |

\* Antigorite: 3643 = (3632 + 3654)/2, the peaks at 3632 and 3654 cm$^{-1}$ are all due to the outer OH sloping stretching vibration.

**Table 6.** The major NIR bands in the range 7000–7600 cm$^{-1}$ related to the serpentine samples and their corresponding MIR peaks (in cm$^{-1}$).

| Lizardite | | | | Chrysotile | | | | Antigorite | | | |
|---|---|---|---|---|---|---|---|---|---|---|---|
| | Measured Peak | Fundamental Peaks | Factor | | Measured peak | Fundamental Peaks | Factor | | Measured Peak | Fundamental Peaks | Factor |
| DWH14-lz | 7170 | 3650 | 1.9644 | SM-ctl | 7170 | 3648 | 1.9655 | H7-atg | 7168 | 3643 * | 1.9676 |
| | 7204 | (3650 + 3689)/2 | 1.9632 | | 7204 | (3648 + 3689)/2 | 1.9637 | | 7204 | (3643 * + 3693)/2 | 1.9640 |
| | 7234 | (3678 + 3689)/2 | 1.9639 | | 7233 | (3668 + 3689)/2 | 1.9663 | | 7231 | (3674 + 3693)/2 | 1.9631 |
| | 7241 | 3689 | 1.9629 | | 7243 | 3689 | 1.9634 | | 7243 | 3693 | 1.9613 |
| ZH41-lz | 7168 | 3647 | 1.9655 | | | | | | 7167 | 3643 * | 1.9673 |
| | 7203 | (3647 + 3688)/2 | 1.9640 | | | | | H11-atg | 7203 | (3643 * + 3696)/2 | 1.9629 |
| | 7233 | (3665 + 3688)/2 | 1.9674 | | | | | | 7230 | (3674 + 3696)/2 | 1.9673 |
| | 7241 | 3688 | 1.9634 | | | | | | 7241 | 3696 | 1.9673 |
| Z3-lz | 7170 | 3649 | 1.9649 | | | | | | | | |
| | 7204 | (3649 + 3688)/2 | 1.9637 | | | | | | | | |
| | 7233 | (3670 + 3688)/2 | 1.9660 | | | | | | | | |
| | 7242 | 3688 | 1.9637 | | | | | | | | |
| SZ17-lz | 7167 | 3648 | 1.9646 | | | | | | | | |
| | 7204 | (3648 + 3688)/2 | 1.9640 | | | | | | | | |
| | 7231 | (3667 + 3688)/2 | 1.9663 | | | | | | | | |
| | 7241 | 3688 | 1.9634 | | | | | | | | |
| | | Average | 1.9645 | | | Average | 1.9647 | | | Average | 1.9651 |

\* Antigorite: 3643 = (3632+3654)/2, the peaks at 3632 and 3654 cm$^{-1}$ are all due to the outer OH sloping stretching vibration.

## 5. Conclusions

1. There are obvious differences in the infrared spectrum of the three serpentine minerals. Lizardite has two equal-height peaks at 4280 and 4300 cm$^{-1}$, and two strong peaks at 7170 and 7241 cm$^{-1}$. Chrysotile has a shoulder peak at 4280 cm$^{-1}$ and 7243 cm$^{-1}$ and only one strong peak at 7233 cm$^{-1}$. There are single strong peaks at 3674 cm$^{-1}$, 4300 cm$^{-1}$, and 7231 cm$^{-1}$ and weak shoulder peaks at 4280 cm$^{-1}$ and 4315 cm$^{-1}$ for antigorite, with a clear peak at 7242 cm$^{-1}$. These characteristic peaks are helpful in identifying and distinguishing between serpentine minerals.

2. In the range 4000–4600 cm$^{-1}$, the serpentine peaks correspond to the OH secondary combination band region and in the range between 4280 and 4300 cm$^{-1}$, it corresponds to a combination of sloping, vertical stretching, and bending vibrations of the outer OH groups. The 7000–7600 cm$^{-1}$ band is the first fundamental overtone of the OH group stretching vibrations. The peaks at 7170, 7204, 7233, and 7242 cm$^{-1}$ correspond to the overtone of the sloping stretching vibration of the outer OH group, the overtone combined sloping stretching vibration and vertical stretching vibration of the outer OH group, and the overtone of the vertical stretching vibration of the outer OH group, respectively.

3. Due to the non-ideal conditions, the actual position of the overtone peak is lower than the theoretical position. The factor of the first fundamental overtone of the OH group stretching vibration is about 1.965.

**Author Contributions:** Conceptualization, M.H.; Data curation, S.W. and B.Z.; Formal analysis, S.W., M.Y. and B.Z.; Funding acquisition, F.W. and Q.L.; Investigation, Q.L.; Methodology, S.W. and M.Y.; Project administration, M.H.; Resources, F.W. and Q.L.; Writing—original draft, S.W. and B.Z.; Writing—review & editing, M.H. and M.Y. All authors have read and agreed to the published version of the manuscript.

**Funding:** National Mineral Rock and Fossil Specimens Resource Center.

**Data Availability Statement:** Not applicable.

**Conflicts of Interest:** The authors declare no conflict of interest.

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
