# Peer review of "Near-Infrared Spectroscopy Study of Serpentine Minerals and Assignment of the OH Group"

_crystals, doi:10.3390/cryst11091130_

Round 1
Reviewer 1 Report
The manuscript I reviewed is, in general, well done. The authors combine two conventional techniques (X-ray diffraction and MIR spectroscopy) with the more uncommon technique (NIR) for mineral characterization. I really appreciate this approach, which allow to better understand the OH group arrangement inside serpentine minerals crystal structure. Of course a similar study could be of interest and I think the approach is absolutely correct. I have no concerns about data acquisition and interpretation. Nevertheless, I think this work is quite lacking in the outcomes. Namely, the Authors analyse three samples of three different -almost pure- phases and perform an analysis of their spectra, but it is not clear the novelty of their approach: in fact spectra interpretation is mainly based on literature data. I think these data could be a good starting point for a future research, but it needs to be better developed, adding more analyses or higher number of samples.
E.g., at L.166 the authors reported that some bands could be due to water trapped in the grains. Well, the interpretation is from another work and the authors don’t investigate if it is true. I suggest to dry the samples in the oven at 105°, evaluate the water loss in weight, and make a comparison of the MIR and NIR spectra before and after treatment.
I have no particular concerns about language. I found the text correct and easy to read. I found some incongruences in the literature notes and some losses the final bibliography. I suggest the authors to check it.
I also found some other things to be fixed reported here below:
L.45: Petriglieri et al., 2015
L.51: Farmer (1974). This citation is lost in References.
L.82: “Three typical mineral samples were selected from the 82 three groups for this study and were labeled H7, SM, and DWH-14.” This sentence is introductive for samples description. In my opinion it should be moved at the beginning of the paragraph. I also suggest the authors to explain the meaning of the samples names in order to help the reader to easily remember them, otherwise they could add a small table with all the samples information (name, provenance, description, composition, reflective index and gravity).
L.106-114: The data description is fine, but in Fig. 1 the scale is in degrees, while in the text the authors refer to XRD peaks in d spacing. I suggest the authors to change the scale or, conversely, to add a note to each peak in order to help the reader to find the peaks in the picture.
L.119: Fig 1 – I suggest the authors to show the raw FTIR spectra in this figure in order to better evaluate the background
L.137: (Kloprogge, 1999; Trittschack, 2012; Post, 2000; Yariv, 1986): check all the citations along the text: sometimes they are reported in this way, sometimes with “et al.” notation. It is not a mistake but the notation should be coherent in all the text.
L.189: King and Clark, 1989; Uehara and Shirozu, 1985: See comments above, these citations are lost.
L.217-219: This observation is correct. Fe-Mg substitution is quite common in phyllosilicate minerals. Anyway, the authors should discuss why it is evident only in but it could be easily verified with a chemical analysis on all the three samples.
Author Response
Dear reviewer,
I made some modifications to the article under the guidance of Professor He Mingyue (The corresponding author). The amount of samples and data is increased, so that the reliability of the conclusion is improved(The conclusion remains unchanged).
Point 1:the Authors analyse three samples of three different -almost pure- phases and perform an analysis of their spectra, but it is not clear the novelty of their approach: in fact spectra interpretation is mainly based on literature data. I think these data could be a good starting point for a future research, but it needs to be better developed, adding more analyses or higher number of samples.
Response 1: The novelty of this paper is to select three kinds of serpentine to compare the spectral differences in detail. Pure samples without any other phase may reduce the impact on the spectrum, and distinguish and explain the similarities and differences of serpentine subspecies. Previous literatures mostly compared the differences of different kinds of layer silicate minerals, and did not distinguish serpentine minerals. Of course, to explain the attribution of spectral peaks, we need to summarize their literature data.
Point 2: E.g., at L.166 the authors reported that some bands could be due to water trapped in the grains. Well, the interpretation is from another work and the authors don’t investigate if it is true. I suggest to dry the samples in the oven at 105°, evaluate the water loss in weight, and make a comparison of the MIR and NIR spectra before and after treatment.
Response 2: Sorry, I didn't describe it in the experiment part. All samples are dried before testing, but the reaction of interlayer water is inevitable. Since interlayer water is not the focus of the study, it is not further discussed. I find out possible explanations from the literature and analyze them in combination with my results.
Of course, your suggestion is of great value, I can further dry the experiment at a higher temperature, but at present, the weather is humid in this season, which has a great impact on the results, the conditions are limited, and the waiting time is very long. Do I have to supplement this experiment?
Point 3: L.217-219: This observation is correct. Fe-Mg substitution is quite common in phyllosilicate minerals. Anyway, the authors should discuss why it is evident only in but it could be easily verified with a chemical analysis on all the three samples.
Response 3: EDS experimental results are added to the sample part. Isomorphism is not the main object of discussion in this manuscript, and fine component analysis will be carried out in future research.
•some other things to be fixed reported here below.
Response 4: other problems such as language, lack of literature, etc. have been corrected / added in the manuscript.
thank you!
Reviewer 2 Report
The submitted manuscript presents results of an analytical study of three minerals (polimorphs) of the serpentine (sub)group by infrared spectroscopy. The data reported will be useful as a reference for future studies on similar minerals. The manuscript is well-written; however, I provide general comments and suggestions that may help improving it.
General comments
- The abstract should be a short summary of the work presented. I suggest changing the abstract to a more general and short text. In particular, the description of the results is probably too detailed for this section.
- Since the study focuses on the application of near and mid IR spectrometry, perhaps in the introduction it is worth expanding the description of the two ranges a little more. A short paragraph on the strengths of both NIR and Mid-IR may add value to the manuscript.
- The bibliographical review of the IR application on serpentine is quite limited. I suggest adding some references (e.g., Groppo et al., 2006; Luce, 1971). Also, a short description of the minerals, highlighting the difference in their structure, may be useful for a reader who may be unfamiliar with the different polymorphs. See for example Demichelis et al., 2016.
- The sample references do not help to identify the polymorph being referred to throughout the text. I suggest changing the labels as: H7-atg for antigorite, SM-ctl for chrysotile, DWH14-lz for lizardite. Please, specify the reference for each mineral at line 83 and change them throughout the manuscript.
- At lines 169-170, you claim that the bands in the ranges 5100–5500 cm-1 and 7000–7500 cm-1 are due to the presence of water, ruling out the possibility that impurities are associated to those bands because their content does not influence the peak position. However, the presence of different phases is not described anywhere in the text. To the contrary, at lines 114 and 126, it is reported that the minerals are almost pure but no further explanation and/or data are given. Please, clarify this point.
Minor comments:
Line 37. I suggest changing “water or other functional groups” with “water and different functional groups”
Line 43. Please, change “layers” for “sheets”.
Line 71. I suggest adding “Finally,” before “Careful measurements”.
Line 75-77. Please, delete these lines. The same information is given in the next paragraph.
Line 79. Please, add “samples” after “lizardite” to avoid misinterpretation.
Line 82. I suggest deleting “typical”.
Line 103. Please, change “the exact locations of the peaks” with “the exact locations of their component peaks”.
Line 141. Please, correct “BISHOP” for “Bishop”.
Line 180: What does “before” refer to? Please, clarify.
Line 183. I suggest adding “It is worth noting that” after the dot.
Line 193. As there are not further subtitles, I would delete this line.
Line 254: Please, specify the band range you refer to.
References
Groppo, C., Rinaudo, C., Cairo, S., Gastaldi, D., & Compagnoni, R. (2006). Micro-Raman spectroscopy for a quick and reliable identification of serpentine minerals from ultramafics. European Journal of Mineralogy, 18(3), 319-329.
Luce, R. W. (1971). Identification of serpentine varieties by infrared absorption. Geol. Sur. Prof. Paper, 750, 199-201.
Demichelis, R., De La Pierre, M., Mookherjee, M., Zicovich-Wilson, C. M., & Orlando, R. (2016). Serpentine polymorphism: a quantitative insight from first-principles calculations. CrystEngComm, 18(23), 4412-4419.
Author Response
Dear reviewer,
I made some modifications to the article under the guidance of Professor He Mingyue (The corresponding author). The amount of samples and data is increased, so that the reliability of the conclusion is improved(The conclusion remains unchanged).
Point 1:The abstract should be a short summary of the work presented. I suggest changing the abstract to a more general and short text. In particular, the description of the results is probably too detailed for this section.
Response 1: I hope to present the main results to the readers, and I prefer not to change them if they are not necessary.
Point 2:Since the study focuses on the application of near and mid IR spectrometry, perhaps in the introduction it is worth expanding the description of the two ranges a little more. A short paragraph on the strengths of both NIR and Mid-IR may add value to the manuscript.
Response 2: I accept your suggestion. The manuscript has been corrected (Line 34-40).
Point 3:The bibliographical review of the IR application on serpentine is quite limited. I suggest adding some references (e.g., Groppo et al., 2006; Luce, 1971). Also, a short description of the minerals, highlighting the difference in their structure, may be useful for a reader who may be unfamiliar with the different polymorphs. See for example Demichelis et al., 2016.
Response 3: I accept your suggestion. The manuscript has been corrected (Line 57-60, 65, Table 2).
Point 4:The sample references do not help to identify the polymorph being referred to throughout the text. I suggest changing the labels as: H7-atg for antigorite, SM-ctl for chrysotile, DWH14-lz for lizardite. Please, specify the reference for each mineral at line 83 and change them throughout the manuscript.
Response 4: I accept your suggestion. The manuscript has been corrected
Point 5:you claim that the bands in the ranges 5100–5500 cm-1 and 7000–7500 cm-1 are due to the presence of water, ruling out the possibility that impurities are associated to those bands because their content does not influence the peak position. However, the presence of different phases is not described anywhere in the text. To the contrary, it is reported that the minerals are almost pure but no further explanation and/or data are given. Please, clarify this point.
Response 5: The "impurity" here may be my wrong expression. I want to express that "the impurity ions replaced by isomorphism in serpentine minerals have no effect on sections 5000-5500 and 7000-7500 cm-1", rather than other mineral phases in the sample. For the content of isomorphic substitution, I supplemented EDS in the sample part.
•Minor comments.
Response 6: other problems such as language, lack of literature, etc. have been corrected / added in the manuscript.
thank you!
Round 2
Reviewer 2 Report
- Table 1 does not show the mineral's composition but only general characteristics and minor elements.
- I would not refer to isomorphic substitutions as impurity. Maybe add quotes ("impurity")
Line 114. Please, correct "datas were were" with "data were"
Author Response
Thank you for your suggestion. I have corrected it. This change has been marked in green (The yellow part is the last modification).
Point 1: Table 1 does not show the mineral's composition but only general characteristics and minor elements.
Response 1: I have changed the order of "sample" and "mineral" columns in Table 1, so that the mineral composition can be showed more clearly.
Point 2: I would not refer to isomorphic substitutions as impurity. Maybe add quotes ("impurity")
Response 2: It has been changed (Line 97-98, 194).
Point 3: Line 114. Please, correct "datas were were" with "data were"
Response 3: It has been changed (Line 114), just a clerical error.